# Rural Work and Specialty Choices of International Students Graduating from Australian Medical Schools: Implications for Policy

**DOI:** 10.3390/ijerph16245056

**Published:** 2019-12-11

**Authors:** Matthew R. McGrail, Belinda G. O’Sullivan, Deborah J. Russell

**Affiliations:** 1Rural Clinical School, The University of Queensland, Rockhampton 4700, Australia; belinda.osullivan@uq.edu.au; 2School of Rural Health, Monash University, Bendigo, 3550, Australia; 3Northern Territory Medical Program, Flinders University, Darwin 800, Australia; deborah.russell@menzies.edu.au; 4Menzies School of Health Research, Darwin 800, Australia

**Keywords:** rural, medical workforce, health policy, international students, maldistribution, general practice, Australia, access

## Abstract

Almost 500 international students graduate from Australian medical schools annually, with around 70% commencing medical work in Australia. If these Foreign Graduates of Accredited Medical Schools (FGAMS) wish to access Medicare benefits, they must initially work in Distribution Priority Areas (mainly rural). This study describes and compares the geographic and specialty distribution of FGAMS. Participants were 18,093 doctors responding to Medicine in Australia: Balancing Employment and Life national annual surveys, 2012–2017. Multiple logistic regression models explored location and specialty outcomes for three training groups (FGAMS; other Australian-trained (domestic) medical graduates (DMGs); and overseas-trained doctors (OTDs)). Only 19% of FGAMS worked rurally, whereas 29% of Australia’s population lives rurally. FGAMS had similar odds of working rurally as DMGs (OR 0.93, 0.77–1.13) and about half the odds of OTDs (OR 0.48, 0.39–0.59). FGAMS were more likely than DMGs to work as general practitioners (GPs) (OR 1.27, 1.03–1.57), but less likely than OTDs (OR 0.74, 0.59–0.92). The distribution of FGAMS, particularly geographically, is sub-optimal for improving Australia’s national medical workforce goals of adequate rural and generalist distribution. Opportunities remain for policy makers to expand current policies and develop a more comprehensive set of levers to promote rural and GP distribution from this group.

## 1. Introduction

Australia’s medical workforce has seen a rapid growth of doctors trained in the last 15 years, with increases in both domestic and international student numbers [1]. Foreign Graduates of Accredited Medical Schools (FGAMS)—defined by the Australian Government Department of Health as “Graduates who received their primary medical qualification from an accredited medical school in Australia or New Zealand, and were not a permanent resident or citizen of either Australia or New Zealand at the time of enrolment”—comprised around 17% (642/3853) of the Australian commencing medical school cohort in 2017. [2] The largest source countries for FGAMS are Singapore (32%), Canada (21%) and Malaysia (12%), with all remaining countries comprising less than 5% of the international medical student intake [1]. There is no government-imposed regulatory cap on full fee-paying international student places or fees, which is in contrast to the cap on both the number of Commonwealth-supported medical school places and fees for students in those places. The private fees international students pay to train in Australian medical schools therefore help to cross-subsidise training of domestic medical students. Universities have control over international student places, albeit limited by their available training resources [3]. Most international students are on student visas until graduation, but opportunities exist for newly qualified FGAMS to remain in Australia by applying for a temporary graduate visa (currently subclass 485—post-study work stream), which allows them and their families to stay in Australia to gain work experience for a period of up to 4 years, with further (including permanent) visa options available after that time [4].

In Australia, FGAMS are not guaranteed internship (first year post-graduation) positions. While each state and territory has its own intern application processes, in general, matching of domestic medical graduates (DMGs) who are permanent resident/citizens occurs first. Subsequently, FGAMS are matched with any remaining places. State and Commonwealth funding has also ensured the expansion of private intern positions since 2013, specifically to accommodate demand from FGAMS [5,6,7,8]. Hawthorne et al., in 2010, estimated that up to 70% of Australia’s FGAMS initially remain working in Australia [9]. Department of Health data, from 2009 to 2017, indicate that 62%–83% of FGAMS successfully gain an intern position (see Table 1). Overall, an average of 348 new FGAMS annually filled an average of almost 3000 intern positions nationally (11.6% of positions). Getting an intern position is an essential step for FGAMS to work as a doctor in Australia, because satisfactory completion of an intern year is essential for general medical registration. General medical registration, in turn, supports applications for other working visa types, pathways to permanent residency and applying for ongoing specialty training—each of which may be very attractive to FGAMS [4].

It is vital for policy makers to understand not only the importance of FGAMS in Australia’s international medical education market, but also the contribution of FGAMS to local workforce goals. Australia is currently developing a national medical workforce strategy, with the key objectives being to reduce geographic maldistribution and address the under-supply of doctors in some specialties [11]. However, the contributions of FGAMS to the rural workforce and general practice specialty is largely invisible [12]. Australian legislation imposes the same regulatory restrictions limiting access to Medicare provider numbers on FGAMS as it does on overseas trained doctors (OTDs)—doctors who received their basic medical degree outside of Australia and New Zealand [13]. This imposes a requirement that FGAMS (and OTDs) work in Distribution Priority Areas (DPA), which are mainly rural areas, in order to access a provider number which is needed for private consultations to be billed to Medicare [14]. This requirement is in place for up to ten years. Notably, doctors practicing in public hospital roles do not need a provider number, so can work in locations that are not DPAs. Evidence suggests that the policy restricting provider number access is associated with high proportions of OTDs working in rural and remote locations, though the effect is not known for FGAMS [15]. Since FGAMS were full-fee students, they often graduate with a large financial debt from their medical training, with evidence suggesting a link between higher student debt and reduced odds of both practicing rurally and working as a GP [16,17,18]. Also, FGAMS, accustomed to the Australian health care system may be better positioned than OTDs to gain jobs in preferred locations (including internships) in the public hospital system when they first begin working in Australia. It is also possible that existing provider number legislation unintentionally deters FGAMS from choosing community medical practice career pathways such as general practice, instead opting to remain working in hospitals and non-GP specialties, to avoid having to work in DPAs. 

To date, little published evidence has described the distribution outcomes of FGAMS. National-level graduate intention data found that FGAMS were significantly more likely to prefer urban than rural practice (OR 1.79, CI 1.19–2.72) [16]. A 2017 study of FGAMS who had graduated in Tasmania, a wholly rural state, found that 33% were working rurally 1–15 years post-graduation). [19] However, no Tasmanian or other comparison data were applied to this study. Evidence from Victoria found that FGAMS had an odds ratio of 5.8 (95% CI 4.0–8.4) for working rurally 1–9 years post-graduation, compared to DMGs [20]. A study of GPs suggested FGAMS had a lower probability of training on a rural pathway, though their definition of FGAMS was problematic as it included doctors born overseas but who were Australian citizens when they entered medical school, and thus would be classified as DMGs by the Australian government [21]. With regards to specialty outcomes of FGAMS, the only identified evidence related to specialty preference on graduation was that FGAMS were equally likely as DMGs to have general practice as their first preference [16].

In summary, the available evidence has considerable limitations. Out study aims to address these by utilising the Commonwealth definition of FGAMS, observing actual behaviour rather than preferences, collecting data at the national scale rather than single institutions and using relevant comparators. This paper thus aims to describe the geographic and specialty distribution of FGAMS within the Australian medical workforce, compared with domestic graduates (DMGs) and doctors entering the medical workforce from another country (OTDs).

## 2. Materials and Methods

This study used 2012–2017 data (waves 5–10) from the Medicine in Australia: Balancing Employment and Life (MABEL) study. MABEL is a national longitudinal study that collects annual survey data from a panel of doctors, with a regular top-up of recently graduated doctors and OTDs newly registered in Australia. MABEL respondents include doctors at all career stages, across all specialties. The MABEL study commenced in 2008 by inviting the entire medical workforce to participate, and 10,498 doctors (19% of the medical population) completed the initial survey (wave 1). There has subsequently been an annual 70%–80% study retention rate, with annual top-ups of new doctors to the sampling frame, through to the most recently available 2017 data (wave 10). Participants complete questionnaires of around 20–30 minutes duration, either hard copy or online. MABEL was approved by the University of Melbourne Faculty of Business and Economics Human Ethics Advisory Group (Ref. 0709559) and the Monash University Standing Committee on Ethics in Research Involving Humans (Ref. CF07/1102-2007000291).

Two main distribution outcomes were measured. Firstly, geographical distribution was defined using the Modified Monash Model (MMM) classification as metropolitan (MMM-1), large regional (MMM-2) or other rural (MMM 3–7). Some analyses further collapsed this to MMM-1 versus rural (MMM 2–7). Secondly, specialty distribution was defined as general practitioners (GPs) versus all other specialties. Vocationally training doctors were categorised according to their enrolled specialty college. Doctors with no specialty or not enrolled with a specialist training program were omitted from analyses relating to specialty.

Three doctor groups were compared: FGAMS, OTDs and DMGs. These groups were identified using a two-stage process. OTDs were firstly identified using country of qualification data as reported on the MABEL questionnaire, with missing data populated using university qualification data from the Australasian Medical Publishing Company (a national data source used as the MABEL sampling frame). FGAMS were delineated from DMGs by their answer to the question “If you completed your medical degree in Australia, were you an international student (i.e., were you a citizen of a country outside of Australia and New Zealand)?” which was introduced to the MABEL survey in 2012 in wave 5 and repeated annually. Where discrepancies of this response occurred in different years, individuals were coded as their majority response.

Other covariates included: gender (male, female); rural background (whether they spent at least 6 years of their childhood in rural areas); and time in Australian workforce (defined according to the number of years since they had commenced working as a doctor in Australia). Two categories were defined, either 0–15 years (to reflect early-mid career doctors with most yet to complete all specialty training and many having practice location restrictions) or >15 years (to reflect established fully trained doctors, all without practice location restrictions), calculated as the time since they graduated from their basic medical degree (FGAMS and DMGs) or first entered the Australian workforce (OTDs). 

Descriptive statistics were used to analyse the distributional outcomes for (i) wave 5 (2012), the first year available with FGAMS data, (ii) wave 10 (2017), the most recent data available, and (iii) aggregated for waves 5–10 (2012–2017 inclusive). Multiple logistic regression models (adjusted for clustering on the same doctor) were used to measure associations between the above key characteristics and main distribution outcomes (working rurally and working as a GP). Consistent with other similar studies, sampling weights were not used to adjust for survey non-response bias because the demographics of the OTD group (older, male) were too different from the population data upon which dataset weightings were originally calculated [15]. Missing values meant 14.7% of responses were dropped from the regression models. All analyses used Stata SE 15.1 for Windows (Stata Corp, College Station, Texas, USA) and statistical significance was set at *p* < 0.05.

## 3. Results

Between waves 5 and 10 (2012–2017), there were 18,093 different doctors who responded at least once. There were 10,101 respondents to wave 5 and 8520 to wave 10. Across all six waves, there was a total of 58,312 aggregate responses (an average of 3.2 responses per individual). 

FGAMS comprised approximately 4% of respondents to MABEL (see Table 2), with OTDs comprising approximately 19% and the remainder being DMGs. The proportion of female respondents was slightly lower in wave 5 than in wave 10 (45.2% versus 48.7%), while the proportion with a rural background was slightly higher in wave 10 (20.2% versus 21.4%) and the proportion with over 15 years in Australia was also higher in wave 10 (51.6% versus 54.2%). Further stratification of FGAMS data revealed only 11.7% (wave 5) and 9.5% (wave 10) had a rural background, significantly less than for DMGs (22.0%) and IMGs (18.6%). Additionally, the proportion of FGAMS respondents who were in the first 15 years of their career increased between wave 5 and wave 10 from 59% to 64%.

The aggregate data for waves 5–10 revealed that FGAMS had the lowest proportion of respondents working rurally (19.3%). (Table 3) A slightly higher proportion of DMGs (than FGAMS) worked rurally (22.1%) whilst an even higher proportion of OTDs (34.7%) were in rural practice, with substantially more OTDs in ‘other rural’ (smaller population) locations. Analysis by career stage showed that for FGAMS graduating within the last 15 years the proportion of FGAMS in rural practice was considerably higher (24.3%, +12.5% in absolute terms) than for FGAMS graduating >15 years ago (11.8%). A similar pattern was seen amongst OTDs, with the proportion of OTDs in rural practice being substantially higher amongst those entering Australia within the last 15 years (40.3%, +14.3%) than those who had been in Australia >15 years (26.0%). For DMGs, however, the proportion in rural practice was similar for those graduating within 15 years (20.9%) versus >15 years (22.1%).

Aggregate data showed similar proportions of FGAMS (42.7%) and DMGs (39.4%) working in general practice, whereas a considerably higher proportion of OTDs were in general practice (48.6%) (Table 3). The proportions of DMGs and FGAMS working in general practice were lower for those graduating within the last 15 years (37.2% and 38.8% respectively) compared to those who had graduated >15 years ago (40.5% and 46.6% respectively). The proportion of OTDs in general practice was similarly high between those who had entered the Australian workforce within the last 15 years (51.4%) and those who had been working in Australia for >15 years (46.8%).

Table 4 reports the statistical significance and strength of associations between doctor group and distributional outcomes, adjusting for gender, rural origin and career cohort. After adjusting for potential confounders, FGAMS overall were no more likely to work rurally compared with DMGs (OR 0.93, 0.77–1.13), but somewhat more likely than DMGs to work as GPs (OR 1.27, 1.03–1.57). In contrast, FGAMS had substantially lower odds of working rurally (OR 0.48, 0.39–0.59) and of working as a GP (OR 0.74, 0.59–0.92) compared with OTDs. When analyses were further stratified, only FGAMS with a rural background were significantly more likely than DMGs with a metropolitan background to work rurally. Female FGAMS were 30% less likely than male DMGs to work rurally. FGAMS of both genders were 65% more likely to work as GPs than male DMGs, though there is a comparatively low proportion of GPs in the reference group (male DMGs). FGAMS who had graduated >15 years ago were about half as likely (OR 0.53, 0.35–0.80) to work rurally compared with DMGs of the same career stage, but more likely than DMGs of the same career stage to work as a GP (OR 1.46, 1.08–1.97).

## 4. Discussion

This paper presents the first empirical evidence at a national level about the workforce distributional outcomes of FGAMS who remain in Australia after their medical school graduation. Key comparisons of workforce outcomes by geography and specialty are made with both DMGs and OTDs. This is important information, especially given Australia’s national interest in achieving medical workforce self-sufficiency with a geographical distribution and specialty mix of doctors able to provide accessible, cost-effective and sustainable care for both metropolitan and rural populations [11]. Australian universities benefit financially from having full-fee paying international medical students, though little attention has been given to their post-graduation outcomes, in terms of the contribution they make to the country’s workforce. A key finding is that while overall FGAMS are no more likely than DMGs to be working rurally, FGAMS who are >15 years since graduation are only half as likely as DMGs to be working rurally. Further, overall, FGAMS are only slightly more likely to be GPs than DMGs, and substantially less likely to be working rurally or as GPs compared to OTDs.

These patterns of geographical and specialty distribution of FGAMS are evident despite Australia imposing a regulatory policy aiming to influence both FGAMS and OTDs to work in Distribution Priority (mainly rural) Areas for up to 10 years (by otherwise limiting access to private billing numbers through Medicare). The findings suggest that the current regulatory policy has some impact on the geographical distribution of FGAMS, but this is not sustained. Amongst FGAMS and OTDs, absolute proportions working rurally are 12%–14% lower at later career stages (>15 years since graduation/working in Australia), after the 10 year moratorium has expired, compared to during earlier career stages. Restrictions on provider number access may be less effective amongst FGAMS because of greater opportunities for FGAMS to access general registration through local internships, albeit after domestic students are prioritised, and then avoid the DPA-related regulation by remaining in hospital employment and hospital-based specialty training, where provider numbers are not required. FGAMS’ desire to maintain practice location autonomy may also be an unintended driver of specialty choices made by FGAMS, driving them away from general practice. This important finding indicates that achieving a more balanced geographic and specialty distribution of FGAMS and sustaining rural practice in the medium and longer term may require a move away from heavy reliance of regulatory approaches alone, to a more comprehensive package of strategies. 

International evidence points to the need for rural workforce retention strategies to be multi-faceted, targeting education (selection and training), personal and professional support and financial support in addition to regulatory approaches [22]. A substantial body of research, for example, suggests that strategies such as training medical students and young doctors in rural areas is effective and delivery of curriculum through longitudinal integrated clerkships combined with training in regional hospitals is associated with both rural and generalist practice [20]. However, current rural training policies in Australia systematically exclude FGAMS from participating in Rural Clinical Schools (undergraduate training) and instead target Commonwealth-supported (DMG) students only. Competitive processes and priority rounds for allocating intern positions to DMGs tends to leave FGAMS with accepting leftover positions, commonly rural [23]. However, with limited local rural exposure during medical school and a substantially lower likelihood of having had rural exposure during childhood, FGAMS initially have few rural social connections and it is unsurprising that their early-career rural retention appears poor. FGAMS are thus likely to need higher levels of professional and personal support during their rural postgraduate training than DMGs [24].

Despite the long history of FGAMS training and working in the Australian health system, government policy around this group remains vague and there is an absence of government regulation of their numbers. With the advent of a new National Medical Workforce Strategy, it is timely to reconsider what the optimal numbers of FGAMS are to enter Australia’s medical workforce, as well as governance (currently unregulated), distribution and educational/professional support issues. Overall, we found that FGAMS had 27% increased odds of uptake of general practice as DMGs, though 26% decreased odds as OTDs. Given the longer term pattern of smaller proportions of DMGs choosing general practice [12], this is positive given that FGAMS are trained in the same system as DMGs, but still concerning for future self-sufficiency of the general practice workforce. 

A key strength of this study is that it provides national-level evidence of a key, but often overlooked, part of Australia’s medical workforce: FGAMS. Most published evidence is not able to identify and thus stratify results for FGAMS, whereas this is a strength of the MABEL study infrastructure and this study. Published graduate workforce outcomes usually pertain to a mix of DMGs and FGAMS. Importantly, this study uses the same definition (FGAMS) as utilised in national health policy, which means these findings can directly inform Australian policy. A limitation, however, is that this definition may vary in other locations which may limit generalisability to other countries. A further limitation of this study is that observations are restricted to 2012–2017, though aggregating 6 years’ data stabilised observed proportions. As a cross-sectional study, only associations rather than causality can be identified. Further, only measured confounders captured by the MABEL data could be accounted for in analyses.

## 5. Conclusions

The number of international students in Australian undergraduate medical programs is substantial, with nearly three-quarters ending up practicing in the Australian medical workforce through internships. Geographic workforce distribution outcomes of FGAMS are very similar to those of other Australian-trained graduates (DMGs). In contrast, FGAMS are more likely to be GPs, though this is mostly attributable to those who have worked >15 years. FGAMS, however, are substantially less likely to be working rurally or as GPs compared to OTDs, despite being subject to the same national policy restricting provider number access as OTDs. Given the number of FGAMS training domestically and staying in Australia following graduation, it is important to consider a comprehensive and connected national policy approach, moving beyond the current single regulatory measure, to promote improved rural location and GP specialty choices.

## Figures and Tables

**Table 1 ijerph-16-05056-t001:** Summary of international medical student graduation and internship (FGAMS) counts for Australia. Most intern positions are filled by graduates from the preceding year, and this table is structured to reflect this connection.

	Number of Medical Graduates					Number of Intern Positions		
Year of Graduation	International Medical Student Graduates	Australian Citizen Medical Graduates	Total Medical Graduates	% International	Year of Medical Internship	Total Intern Positions Funded	Total Number of FGAMS with Intern Positions *	Approximate Proportion of FGAMS Accepting Intern Positions
1999–2002 (average)	143	1230	1372	10.4%	2000–2003	n/a	n/a	-
2003–2007 (average)	258	1350	1608	15.9%	2004–2008 (average)	1746	n/a	-
2008	401	1738	2139	18.7%	2009	2243	320	80%
2009	465	1915	2380	19.5%	2010	2394	386	83%
2010	474	2259	2733	17.3%	2011	2723	390	82%
2011	457	2507	2964	15.4%	2012	2950	351	77%
2012	507	2777	3284	15.4%	2013	3118	312	62%
2013	497	2944	3441	14.4%	2014	3287	353	71%
2014	469	2968	3437	13.6%	2015	3305	331	71%
2015	492	3055	3547	13.9%	2016	3420	335	68%
2016	484	3085	3569	13.6%	2017	3466	356	74%

FGAMS = Foreign Graduate of Accredited Medical Schools.* From 2013, FGAMS intern positions were a mix of Commonwealth/private (approximately n = 100) and State-funded positions; n/a = Not available. Data sourced from the Australian Government [1,2,10].

**Table 2 ijerph-16-05056-t002:** Baseline characteristics of study participants (wave 5–10).

Variable	Characteristics	Wave 5 (2012)	Wave 10 (2017)	Aggregate Wave 5–Wave 10
Independents				
Doctor type	DMG	7361 (72.9%)	6700 (78.6%)	40,682 (76.8%)
	FGAMS	414 (4.1%)	344 (4.0%)	2191 (4.1%)
	OTD	2326 (23.0%)	1476 (17.3%)	10,133 (19.1%)
Gender	Male	5539 (54.8%)	4374 (51.4%)	27,993 (52.9%)
	Female	4562 (45.2%)	4144 (48.7%)	24,940 (47.1%)
Rural background	Rural Australia/NZ (DMG)	1482 (15.5%)	1410 (18.1%)	8376 (17.0%)
	Rural elsewhere (FGAMS)	46 (0.5%)	30 (0.4%)	216 (0.4%)
	Rural elsewhere (OTD)	399 (4.2%)	227 (2.9%)	1719 (3.5%)
	Not rural	7623 (79.8%)	6125 (78.6%)	38,965 (79.1%)
Time in Australian workforce	0–15 years	4792 (48.4%)	3870 (45.8%)	24,377 (46.6%)
	>15 years	5112 (51.6%)	4584 (54.2%)	27,972 (53.4%)
Outcomes				
Geographical distribution	Metropolitan	7608 (75.7%)	6312 (74.8%)	39,773 (75.6%)
	Large regional	1060 (10.6%)	890 (10.6%)	5393 (10.3%)
	Other rural	1382 (13.8%)	1238 (14.7%)	7439 (14.1%)
Specialty #	General practice	3361 (41%)	3143 (42%)	18,924 (41%)
	All other specialties	4772 (59%)	4414 (58%)	26,726 (59%)
TOTAL		10,101	8520	53,006

FGAMS = Foreign Graduate of Accredited Medical Schools; OTD = Overseas Trained Doctor; DMG = Domestic Medical Graduate. # Doctors with no specialty or not enrolled with a specialist training program were omitted from this aspect.

**Table 3 ijerph-16-05056-t003:** Rural and specialty distribution by FGAMS compared with DMG and OTDs, by career stage.

Career Stage	Doctor Group	Work Rurality			Specialty #	
		Metropolitan	Large Regional	Other Rural	General Practice	All Other Specialties
Wave 5–10: 0–15 years in Australian workforce	DMGs (17,011)	79.1%	10.0%	10.9%	37.2%	62.8%
	FGAMS (1313)	75.7%	11.6%	12.6%	38.8%	61.2%
	OTDs (6053)	59.7%	14.2%	26.1%	51.4%	48.6%
Wave 5–10: >15 years in Australian workforce	DMGs (23,456)	77.2%	9.6%	13.3%	40.5%	59.5%
	FGAMS (857)	88.2%	2.3%	9.5%	46.6%	53.4%
	OTDs (3659)	74.0%	9.9%	16.1%	46.8%	53.2%
Aggregate Wave 5–10	DMGs (40,467)	77.9%	9.8%	12.3%	39.4%	60.6%
	FGAMS (2170)	80.7%	8.0%	11.3%	42.7%	57.3%
	OTDs (9710)	65.3%	12.7%	22.0%	48.6%	51.4%

FGAMS = Foreign Graduate of Accredited Medical Schools; OTD = Overseas Trained Doctor; DMG = Domestic Medical Graduate. # Doctors with no specialty or not enrolled with a specialist training program were omitted from this aspect.

**Table 4 ijerph-16-05056-t004:** Multivariate logistic regression models of rural and specialty distribution outcomes by FGAMS compared with other groups.

Reference Category	Doctor Characteristics	Work as a Rural Doctor (OR, 95% CI)	Work as a GP # (OR, 95% CI)
Model 1:			
Ref: DMG	FGAMS	0.93 (0.77–1.13)	1.27 (1.03–1.57) *
Ref: OTDs	FGAMS	0.48 (0.39–0.59) **	0.74 (0.59–0.92) **
Ref: Male	Female	0.81 (0.74–0.88) **	1.69 (1.55–1.83) **
Ref: Metro origin	Rural origin	2.54 (2.32–2.79) **	1.30 (1.18–1.44) **
Ref: >15 years work in Aus	0–15 years work in Aus	1.05 (0.97–1.13)	0.79 (0.73–0.86) **
Model 2: Doctor Group and Childhood Origin Ref: DMG and Metro BG	DMG and Rural BG	2.73 (2.46–3.03) **	1.29 (1.16–1.44) **
	OTD and Metro BG	2.11 (1.89–2.35) **	1.71 (1.53–1.91) **
	OTD and Rural BG	3.96 (3.25–4.81) **	2.32 (1.88–2.87) **
	FGAMS and Metro BG	0.94 (0.76–1.16)	1.27 (1.02–1.59) *
	FGAMS and Rural BG	2.64 (1.63–4.28) **	1.60 (0.84–3.03)
Model 3: Doctor Group and Gender Ref: DMG and Male	DMG and Female	0.83 (0.75–0.91) **	1.77 (1.61–1.95) **
	OTD and Male	2.00 (1.76–2.27) **	1.84 (1.61–2.09) **
	OTD and Female	1.52 (1.30–1.77) **	2.82 (2.41–3.30) **
	FGAMS and Male	1.00 (0.77–1.30)	1.65 (1.25–2.17) **
	FGAMS and Female	0.70 (0.53–0.93) *	1.65 (1.22–2.23) **
Model 4: Doctor Group and Work time in Australia Ref: DMG and >15 years	DMG and 0–15 years	0.84 (0.76–0.92) **	0.72 (0.66–0.79) **
	OTD and >15 years	1.16 (0.98–1.38)	1.35 (1.16–1.57) **
	OTD and 0–15 years	2.35 (2.09–2.64) **	1.54 (1.37–1.73) **
	FGAMS and >15 years	0.53 (0.35–0.80) **	1.46 (1.08–1.97) *
	FGAMS and 0–15 years	1.15 (0.94–1.42)	0.82 (0.62–1.08)

FGAMS = Foreign Graduate of Accredited Medical Schools; OTD = Overseas Trained Doctor; DMG = Domestic Medical Graduate; OR = Odds Ratio; CI = confidence interval; * *p*-value <0.05; ** *p*-value < 0.01. Models accounted for clustering of repeated measures on individual doctors. # Doctors with no specialty or not enrolled with a specialist training program were omitted from this aspect.

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
