# Peer review of "Rural Work and Specialty Choices of International Students Graduating from Australian Medical Schools: Implications for Policy"

_ijerph, 2019, doi:10.3390/ijerph16245056_

Round 1

Reviewer 1 Report

This paper adds an important perspective to the topic of engaging medical graduates in rural practice in Australia. 

The topic is introduced in a clear and compelling manner, highlighting  relevant

contextualising information and identifying the need for this study. 

Table 1 is a bit confusing, with the two sections not aligning and the top 9 rows of the intern position section being unnecessary. I suggest a revision of this content to present it in a clearer format. 

The use of the data from the MABEL survey is appropriate and clearly described.  

The methods and results are presented well with clear explanations of steps and findings. 

The discussion is well structured and coherent, raising a number of very relevant findings for medical workforce in rural Australia. The discussion about the education debts of the various doctor groups is interesting and relevant, but should perhaps be introduced in as a concept earlier in the paper and the data presented from the MABEL survey, rather than an additional item in the discussion to strengthen the understanding of this point. Or alternatively you could leave it out of the discussion. The conclusions are in keeping with the strength of the findings and provide an important insight into rural practice outcomes. 

Author Response

Comment 1

This paper adds an important perspective to the topic of engaging medical graduates in rural practice in Australia. 

The topic is introduced in a clear and compelling manner, highlighting relevant contextualising information and identifying the need for this study. 

The use of the data from the MABEL survey is appropriate and clearly described.  

The methods and results are presented well with clear explanations of steps and findings. 

The discussion is well structured and coherent, raising a number of very relevant findings for medical workforce in rural Australia.

The conclusions are in keeping with the strength of the findings and provide an important insight into rural practice outcomes. 

Author response

We thank the reviewer for these positive comments.

Comment 2

Table 1 is a bit confusing, with the two sections not aligning and the top 9 rows of the intern position section being unnecessary. I suggest a revision of this content to present it in a clearer format. 

Author response

The 2 sections in Table 1 are structured correctly but represent on the left side, cohorts of graduating medical students per year of graduation and the right half, interns, which in Australia are those in their 1st year after graduating from medical school. Thus, this table is demonstrating the link between the number of graduates and the number of interns the following year. An extra sentence has been added to the Table header to better clarify this structure.

To simplify the table, we have combined the periods 1999-2002 and 2003-2007 (with average graduate counts - noting that intern position counts were not available for these cohorts).

The first rows (showing averages) are necessary because they provide the baseline for identifying the growth in the number of international medical student graduates between 1999-2002 and 2003-2007, onwards.  They also show the start of the growing number of Australian (citizen) medical graduates and funded intern positions since 2004.  

Comment 3

The discussion about the education debts of the various doctor groups is interesting and relevant, but should perhaps be introduced in as a concept earlier in the paper and the data presented from the MABEL survey, rather than an additional item in the discussion to strengthen the understanding of this point. Or alternatively you could leave it out of the discussion.

Author response

We have considered the issue of education debt and have decided to remove the introduction of new data in the Discussion section. We have adjusted this section by removing a few related sentences.

Reviewer 2 Report

This is a very useful study, well presented.  It might be worthwhile to take another look at all of the data tables for readability and reader comprehension.  They are sufficiently complex that it takes even a well-informed reader time to sort out and think about.

Author Response

Comment 1

This is a very useful study, well presented.  It might be worthwhile to take another look at all of the data tables for readability and reader comprehension.  They are sufficiently complex that it takes even a well-informed reader time to sort out and think about.

Author response

As per comment 2 (reviewer 1), we have adjusted Table 1 to improve its readability. We believe that Tables 2 and 3 do not need editing. Table 4 has had a minor edit, to clarify that 4 separate logistic models were used in producing the results.

Reviewer 3 Report

The asymmetry in the distribution of health professionals across the different regions of the countries is across the western world.

Here is presented the Australian reality.

I congratulate the authors for their excellent analysis of the topic and discussion.

They should correct only the relative values of table 2. In some columns the sum is> 100%.

They should correct only the relative values of table 3. In some columns the sum is> and <100%.

Author Response

Comment 1

The asymmetry in the distribution of health professionals across the different regions of the countries is across the western world. Here is presented the Australian reality.

I congratulate the authors for their excellent analysis of the topic and discussion.

They should correct only the relative values of table 2. In some columns the sum is> 100%.

They should correct only the relative values of table 3. In some columns the sum is> and <100%.

Author response

We have re-checked the percentages in Tables 2 and 3, all of them are correct. The reviewer is correct that a few add to 100.1 and 99.9, rather than exactly 100.0. This is explained due to rounding of numbers to 1 decimal point.